# Particleboards from Recycled Wood

**Ján Iždinský \*, Zuzana Vidholdová[ID] and Ladislav Reinprecht[ID]**

Department of Wood Technology, Faculty of Wood Sciences and Technology, Technical University in Zvolen, T. G. Masaryka 24, 960 01 Zvolen, Slovakia; zuzana.vidholdova@tuzvo.sk (Z.V.); reinprecht@tuzvo.sk (L.R.)

\* Correspondence: jan.izdinsky@tuzvo.sk

**Abstract:** The effective recovery of wood waste generated in wood processing and also at the end of wood product life is important from environmental and economic points of view. In a laboratory, 16 mm-thick three-layer urea–formaldehyde (UF)-bonded particleboards (PBs) were produced at 5.8 MPa and 240 °C and with an 8 s/mm pressing factor, using wood particles prepared from (1) fresh spruce wood (C), (2) a mixture of several recycled wood products (R1), and (3) recycled faulty PBs bonded with UF resin (R2). Particles from spruce wood were combined with particles from R1 or R2 recyclates in weight ratios of 100:0, 80:20, 50:50 and 0:100. In comparison to the control spruce PB, the PBs containing the R1 recyclate from old wood products were characterized by lower thickness swelling after 2 and 24 h (TS-2h and TS-24h), lower by 18 and 31%; water absorption after 2 and 24 h (WA-2h and WA-24h), lower by 33 and 28%; modulus of rupture in bending (MOR), lower by 28%; modulus of elasticity in bending (MOE), lower by 18%; internal bond (IB), lower by 33%; and resistance to decay determined by the mass loss under the action of the brown-rot fungus *Coniophora puteana* (Δm), lower by 32%. The PBs containing the R2 recyclate from faulty PBs were also characterized by a lower TS-2h and TS-24h, lower by 45% and 59%; WA-2h and WA-24h, lower by 61% and 51%; MOR, lower by 37%; MOE, lower by 17%; and IB, lower by 33%; however, their biological resistance to *C. puteana* was more effective, with a decreased Δm in the decay test, lower by 44%.

**Keywords:** particleboards; recycled wood; physical and mechanical properties; decay resistance

## 1. Introduction

Today, manufacturing agglomerated materials, including particleboards (PBs), is based on the idea of utilizing the wood waste of lower value resulting from wood processing, such as sawdust, chips, particles, and wood pulp [1]. An increase in the volume of PB manufacturing has resulted in manufacturers seeking new material resources. At the present time, besides natural wood, other lignocellulose materials [2–4], as well as recycled wood [5,6], are used, especially in countries where there is a scarcity of wood resources [7].

Recycled wood is defined as various kinds of old and residual wood, such as wastes from furniture, construction, etc., and packaging as stated in the Waste Framework Directive (2008/98/EC) [6,8,9]. The increase in the volume of PB manufacturing is associated with PB consumption growth. In 2017, the manufacturing of PBs in Europe was $43.9 \times 10^6$ m$^3$, and in 2018, there was an increase of more than 4.1% [10,11]. Therefore, taking into account the continued growth in the production as well as in the consumption of PBs, it can be assumed that a large amount of materials from PBs will have to be eliminated or recycled each year [12]. Wan et al. [13] mentioned that there have been developed several methods for reconstituting the waste from PBs and other medium density fiberboards (MDF) in order to solve the problem of recycling the wood composites. The processing can be chemically thermomechanical [14,15], hydrothermal [16–18], chemical [19] or mechanical [20–24]. According to Wan et al. [13], it was confirmed by research results that common types of urea–formaldehyde (UF)

and phenol–formaldehyde (PF) resins are suitable for manufacturing PBs and MDF from recycled wood materials using conventional technology.

According to Czarnecki et al. [24], the mentioned methods of wood waste reconstitution are rather difficult to apply in practice, besides the way of mechanical processing when manufacturing PBs. For example, the hydrolytic way of obtaining wood particles from residual PBs can only be used in the case of them being bonded with UF resin.

The wood waste from recycled composite materials should not be found in municipal waste landfills because of its possible degradation products responsible for damaging the environment [25–28]. It should be used as a material for new composites to become more environmentally friendly [25,27,29–34].

When using the recycled wood for composite manufacturing, the decontamination and subsequent grading can be considered a disadvantage. At the present time, several companies possess sophisticated equipment and technology for collecting, grading and decontaminating old timber. Subsequently, it can be used to manufacture PBs [35]. The properties of the new PB can be significantly affected by the recycled wood, owing to the type, size and additives used.

Several research studies aimed at determining the effects of various types of recycled wood and other lignocellulosic waste on the properties of PBs have been conducted across the world. Laskowska and Mamiński [36] studied the properties of PBs manufactured from plywood waste. When manufacturing three-layer PBs, Czarnecki et al. [24] replaced, in the core layer, 10 to 60% of the pine wood particles with particles produced from: (a) three-layer PBs bonded with UF resin, (b) PBs bonded with PF resin, and (c) MDF bonded with UF resin. They found out that the bending strength declined slightly when 10 to 50% particles obtained from recycled PBs bonded with UF resin were added. However, there was a significant decline in the case of 60% recycled particles being added. On the contrary, the internal bond (IB) declined significantly when the portion of recycled particles in the PBs was only 10%. The thickness swelling (TS) of PBs with higher portions of recyclates decreased after 2 as well as 24 h. PBs manufactured from particles obtained from recycled PBs bonded with PF resin showed the worst properties. This could be caused by the negative interaction of the original PF resin and new UF resin used. In this case, only PBs with a minimum of 10% recyclates in the central layer met the requirements of the standard. The best results were obtained using the particles from recycled MDF bonded with UF resin, whose 60% addition to PBs resulted in only a slight decrease in the modulus of rupture (MOR), and even an increase in the IB and TS after 2 and 24 h. Azambuja et al. [37] manufactured PBs using four types of wood recyclates from demolished buildings, i.e., MDF boards, PBs, plywood and timber. In comparison to control PBs from pine particles, the best properties were shown for the PBs manufactured from recycled timber. However, considering the low values of the MOR and modulus of elasticity (MOE), using the wood recyclates only in the core layer of PBs was recommended. Weber and Iwakiri [38] found out that plywood, MDF and PB wastes can be used to manufacture PBs, using them individually or in mixtures. Hameed et al. [39] manufactured three-layered PBs using two types of recycled wood waste material (untreated wood—waste made of massive wood—and slightly treated wood—waste issued from coating or gluing treatments and made of massive wood or other wood-based panels), which were bonded by a rape resin based on leftover cakes of rape oil in a natural state. Both PB types with recycled wood met the requirements of particleboard type P2 (no load-bearing panel for interior use in dry conditions) according to EN 312 [40].

The effects of two types of recycled wood particles – recyclates from old used wood products (R1) and from new but faulty PBs (R2), added in various amounts to spruce particles prepared from freshly cut logs—on selected physical, mechanical and biological properties of PBs were studied for the paper.

## 2. Materials and Methods

### 2.1. Materials

#### 2.1.1. Wood Particles

By the company Kronospan Zvolen, Slovakia, industrial wood chips were prepared from (a) fresh spruce logs (C); (b) a mixture of several recycled wood products consisting of approximately 35% hardboards (HB) and MDF boards, 30% PBs, 20% pallets from spruce wood and 15% old furniture (R1); and (c) recycled faulty PBs bonded with UF resin (R2). All chip types were prepared in the Knife Ring Flakers G24 (GOOS Engineering s.r.o., Slovakia). Subsequently, from the chips were prepared wood particles in the laboratories of the Technical University in Zvolen, using the grinding mill SU 1 (TMS Pardubice, Czech Republic). The dimensions of the particles used in the core layer of the PBs ranged from 0.25 to 4.0 mm and in the surface layers from 0.125 to 1.0 mm. The oversized fraction and the dust were sorted and excluded from the experiment. Fractions falling through a sieve with a mesh diameter of 0.125 mm were considered to be the dust. Subsequently, the particles were dried to a moisture content of 2% in the case of the core layer and to a moisture content of 4% in the case of the surface layers.

#### 2.1.2. Resin and Additives

The UF resin KRONORES CB 4005 D was used in the surface layers, and the UF resin KRONORES CB 1637 D, in the core layer of the PBs (Table 1). The UF resin was added to particles for the surface layers in an amount of 11%, and to particles for the core layer in an amount of 7%. Ammonium nitrate as a 57% water solution (2 or 4% of the dry mass of the UF resin in the case of the surface particles or core particles) was used as a hardener for the UF resins. A paraffin emulsion, with 35% dry mass, was applied on the surface and core particles in amounts of 0.6 and 0.7%, respectively.

**Table 1.** Properties of urea–formaldehyde (UF) resins.

| Quality Parameters | Unit | Method | KRONORES CB 4005 D | KRONORES CB 1637 D |
|---|---|---|---|---|
| Solid content | % | EN 827 [41] | 65.87 | 67.35 |
| Ford cup viscosity, 4 mm/20 °C | s | EN ISO 2431 [42] | 76 | 86 |
| pH value | | EN 1245 [43] | 9.08 | 8.62 |
| Gel time at 100 °C | s | Kronospan chloride test | 81 | 36 |

### 2.2. Particleboard Preparation

Three-layer PBs with dimensions of $400 \times 300 \times 16$ mm and with a density of $650 \pm 10$ kg·m$^{-3}$ were produced under laboratory conditions. UF resin with the hardener was applied on the conditioned wood particles in a laboratory rotary mixing device (VDL, TU Zvolen, Slovakia). The moisture content of the wood particles after applying the adhesive was 9.1 to 10.4% for the surface layers and 6.3 to 7.1% for the core layer. The particle mat was layered manually in wooden forms. The surface/core particle ratio was 35:65. The particle mat was cold pre-pressed in a low-temperature environment at a pressure of 1 MPa, and then, it was pressed in the CBJ 100-11 laboratory press (TOS, Rakovník, Czech Republic). Pressing was conducted in accordance with the pressing diagram (Figure 1)—at a maximum temperature of 240 °C, a maximum pressing pressure of 5.75 MPa, and a pressing factor of 8 s/mm. In total, 42 PBs were manufactured, i.e., 6 from each type (Table 2).

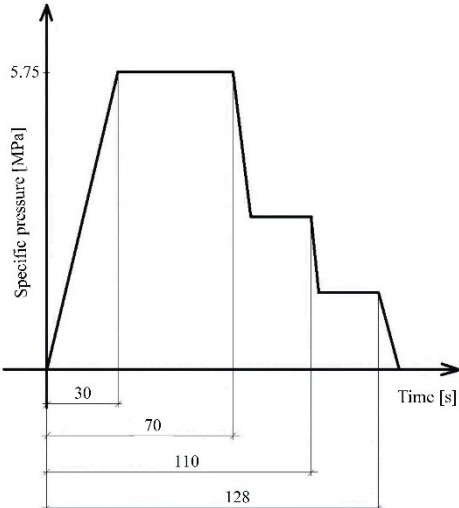

**Figure 1.** Standard three-stage pressing diagram for the manufacturing of particleboards (PBs).

**Table 2.** Individual types of manufactured particleboards (PBs).

| Variant | Amount of Recycled Wood in PB, *w/w* (%) | Number of Produced Boards | Board Type |
|---|---|---|---|
| PB-C: | | | |
| 100% particles of spruce wood | 0 | 6 | C |
| PB-R1: | | | |
| 20%, 50% or 100% particles from | 20 | 6 | 20 R1 |
| mixture of recycled wood products, | 50 | 6 | 50 R1 |
| combined with 80, 50 or 0% | 100 | 6 | 100 R1 |
| particles of spruce wood | | | |
| PB-R2: | | | |
| 20%, 50% or 100% particles from | 20 | 6 | 20 R2 |
| faulty PBs, combined with 80, 50 | 50 | 6 | 50 R2 |
| or 0% particles of spruce wood | 100 | 6 | 100 R2 |

### 2.3. Physical and Mechanical Properties of PBs

The selected properties of the PBs were determined according to the European (EN) standards or Slovak (STN) standards: the density by EN 323 [44], the moisture content by EN 322 [45], the TS and water absorption (WA) after 2 and 24 h by EN 317 [46] and STN 490164 [47], the MOR in bending and the MOE in bending by EN 310 [48], and the IB, i.e., the tensile strength perpendicular to the plane of the PBs, by EN 319 [49]. The samples for these tests were prepared from the 4-week air-conditioned PBs (Figure 2). The universal machine TiraTest 2200 (VEB TIW Rauenstein, Germany) was used to analyze the mechanical properties of the PBs. The classification of the PBs was performed in accordance with the European standard EN 312 [40], taking into account the requirements of the PB (type P2), with a thickness ranging between 13 and 20 mm.

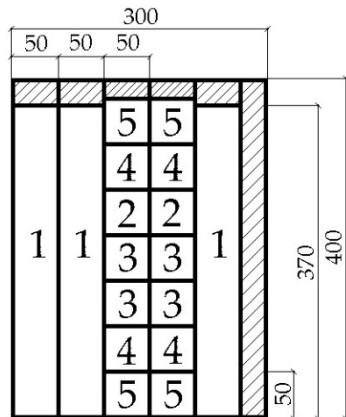

**Figure 2.** Cutting scheme for laboratory-manufactured PBs: 1—samples for testing modulus of rupture (MOR) and modulus of elasticity (MOE) by 3-point bending test [48], 2—samples for testing thickness swelling (TS) and water absorption (WA) after 2 and 24 h [46,47], 3—samples for testing internal bond (IB) [49] and density [44], 4—samples for testing biological resistance to the fungus *Coniophora puteana* [50], and 5—spare samples.

## 2.4. Decay Resistance of PBs

The resistance of all the prepared PBs to the brown-rot fungus *Coniophora puteana* (strain BAM Ebw. 15) was tested according to ENV 12038 [50] using the samples with dimensions of $50 \times 50 \times 16$ mm. Firstly, the samples were conditioned at a temperature of $20 \pm 2$ °C and a relative humidity of $65 \pm 2\%$, achieving an equilibrium moisture content (EMC) of $12 \pm 1\%$. The edges of the samples ($50 \times 16$ mm) were sealed with the epoxy resin CHS-Epoxy 1200 mixed with the hardener P11 in a weight ratio of 11:1 (Stachema, Mělník, Czech Republic), in the amount of $200 \pm 10$ g·m$^{-2}$. Subsequently, the samples were oven dried at rising temperatures from 60 to $103 \pm 2$ °C for 10 h. The sterilized samples were cooled in a desiccator and weighed ($m_0$). Finally, all the surfaces of the PB samples were sterilized twice for 30 min with a UV light radiator. Then, they were soaked in distilled water for 240 min to achieve a moisture content of 25 to 30%. Finally, they were placed into 1 L Kolle flasks on top of the stainless steel grids, with the fungal mycelium inoculant grown on the agar-malt soil (HiMedia Laboratories Pvt. Ltd., Mumbai, India). All the Kolle flasks were incubated for 16 weeks at a temperature of $22 \pm 2$ °C and a relative humidity of 75 to 80%. After the mycological test, the samples of PBs—after depriving the fungal mycelia of their surfaces—underwent a gradual drying process until constant weights ($m_{0/decayed/}$) were achieved, i.e., 100 h at 20 °C, 1 h at 60 °C, 1 h at 80 °C, 8 h at $103 \pm 2$ °C, and final cooling in desiccators. Their mass losses were calculated in percentages using Equation (1):

$$\Delta m = \frac{m_0 - m_{0/decayed/}}{m_0} \cdot 100 \ (\%) \tag{1}$$

## 2.5. Statistical Analyses

The statistical software STATISTICA 12 was used to analyze the gathered data. The descriptive statistics deal with the basic statistical characteristics of the studied properties—the arithmetic mean and standard deviation. Simple linear correlation analysis together with the coefficient of determination was used as a method of inductive statistics to evaluate the measured data.

## 3. Results and Discussion

### 3.1. Physical and Mechanical Properties of PBs

The basic physical and mechanical properties of the PBs manufactured in the laboratory of TU in Zvolen are presented in Tables 3 and 4. The effects of the recyclates R1 and R2 on the properties

of the PBs were analyzed by the linear correlations and the coefficients to determine $r^2$, as shown in Figures 3–7.

The density of the PBs ranged between 649 and 658 kg·m$^{-3}$. It was in a very narrow interval, and it was not apparently affected by the type and amount of wood recyclates (Tables 3 and 4). This finding was also confirmed by the $r^2$ values being lower than 0.01 (Figure 3). This means that the tested moisture, mechanical and biological properties of the PBs could not be affected by density.

**Table 3.** Physical and mechanical properties of the control PB (PB-C) and of the PBs containing particles from a mixture of wood recycled products—R1.

| Property of PB | | Recyclate R1 in PB *w/w* (%) | | | |
|---|---|---|---|---|---|
| | | **0** | **20** | **50** | **100** |
| Density | (kg·m$^{-3}$) | 653 (18.22) | 652 (18.42) | 649 (23.81) | 654 (22.26) |
| Thickness swelling (TS) after 2 h | (%) | 6.05 (0.53) | 4.98 (0.51) | 4.93 (0.56) | 5.65 (1.18) |
| Thickness swelling (TS) after 24 h | (%) | 23.95 (1.39) | 16.57 (2.64) | 22.76 (3.52) | 21.12 (3.83) |
| Water absorption (WA) after 2 h | (%) | 27.59 (2.05) | 18.54 (1.74) | 21.38 (2.42) | 19.55 (3.87) |
| Water absorption (WA) after 24 h | (%) | 68.52 (2.33) | 49.23 (3.95) | 62.02 (6.05) | 54.86 (6.52) |
| Internal bond (IB) | (MPa) | 0.82 (0.06) | 0.75 (0.04) | 0.58 (0.06) | 0.55 (0.06) |
| Modulus of rupture (MOR) | (MPa) | 14.7 (1.57) | 14.7 (1.48) | 11.6 (1.33) | 10.6 (1.40) |
| Modulus of elasticity (MOE) | (MPa) | 2637 (288) | 2666 (246) | 2442 (170) | 2155 (284) |

Notes: Mean values: density from 42 samples, TS from 12 samples, WA from 12 samples, IB from 24 samples, MOR and MOE from 18 samples. Standard deviations are in the parentheses.

**Table 4.** Physical and mechanical properties of the control PB (PB-C) and of the PBs containing particles from recycled PBs—R2.

| Property of PB | | Recyclate R2 in PB *w/w* (%) | | | |
|---|---|---|---|---|---|
| | | **0** | **20** | **50** | **100** |
| Density | (kg·m$^{-3}$) | 653 (18.22) | 658 (16.53) | 652 (20.10) | 652 (21.63) |
| Thickness swelling (TS) after 2 h | (%) | 6.05 (0.53) | 4.48 (0.40) | 4.19 (0.56) | 3.32 (0.48) |
| Thickness swelling (TS) after 24 h | (%) | 23.95 (1.39) | 13.29 (1.35) | 12.43 (1.26) | 9.72 (0.98) |
| Water absorption (WA) after 2 h | (%) | 27.59 (2.05) | 16.41 (0.95) | 15.35 (0.95) | 10.80 (0.63) |
| Water absorption (WA) after 24 h | (%) | 68.52 (2.33) | 43.43 (1.91) | 41.30 (1.51) | 33.46 (1.44) |
| Internal bond (IB) strength | (MPa) | 0.82 (0.06) | 0.74 (0.05) | 0.68 (0.05) | 0.55 (0.12) |
| Modulus of rupture (MOR) | (MPa) | 14.7 (1.57) | 14.4 (1.18) | 11.8 (1.24) | 9.3 (1.27) |
| Modulus of elasticity (MOE) | (MPa) | 2637 (288) | 2800 (166) | 2486 (173) | 2194 (264) |

Notes: Mean values: density from 42 samples, TS from 12 samples, WA from 12 samples, IB from 24 samples, MOR and MOE from 18 samples. Standard deviations are in the parentheses.

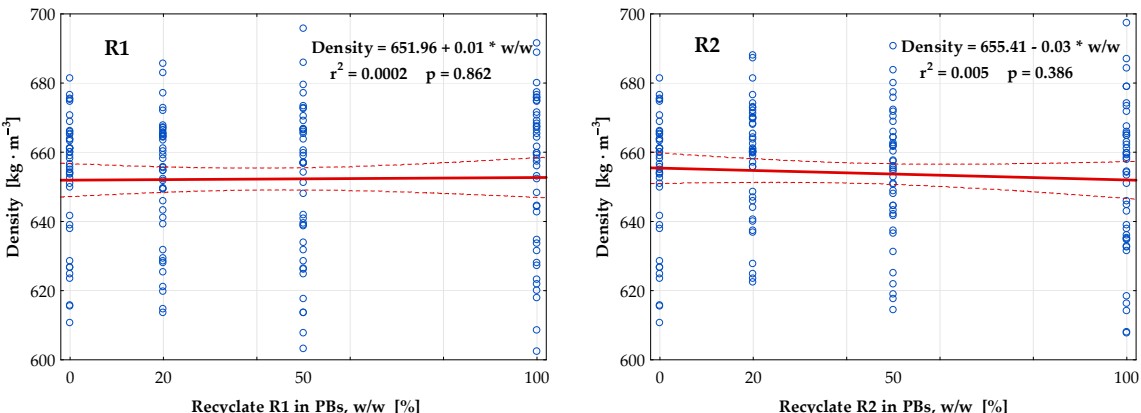

**Figure 3.** Density of PBs containing different types (R1 or R2) and amounts of wood recyclates.

The moisture properties of the PBs, i.e., the thickness swelling (TS) and water absorption (WA), were already more scattered: TS-2h ranged from 3.32% for PB-R2-100 to 6.05% for PB-C; TS-24h ranged from 9.72% for PB-R2-100 to 23.95% for PB-C; WA-2h ranged from 10.80% for PB-R2-100 to 27.59% for PB-C; WA-24h ranged from 33.46% for PB-R2-100 to 68.52% for PB-C (Tables 3 and 4). The effect of the

R1 recycled particles—from the mixture of recycled wood products—on the moisture properties of the PBs cannot be considered as significant, as the $r^2$ value of the linear correlations ranged only between 0.0001 and 0.09. Only in the case of WA-2h was the effect of the R1 particles slightly stronger, as the $r^2$ was 0.22 (Figures 4 and 5). The result that the larger decrease in TS and WA was observed in the case of the PBs that contained only minimal amounts (20%) of the R1 recyclate is interesting—TS-2h decreased by about 18%, TS-24h by about 31%, WA-2h by about 33%, and WA-24h by about 28%. On the contrary, the R1 recycled particles—from the faulty recycled PBs—significantly improved the moisture properties of the laboratory-prepared PBs, the most for PB-R2-100, containing 100% R2 recyclate, with decreases in TS-2h and TS-24h of about 45 and 59%, or in WA-2h and WA-24h of about 61 and 51% (Tables 3 and 4). This result can be explained by the presence of the cured UF resin macromolecules on the surfaces of the R2 particles resulting in a slower transport of water into PBs-R2. Generally, the R2 particles had a significantly positive effect on the moisture properties of the PBs, with the coefficients of determination $r^2$ of the linear correlations "TS or WA = a + b × *w/w*" ranging from 0.63 to 0.72 (Figures 4 and 5).

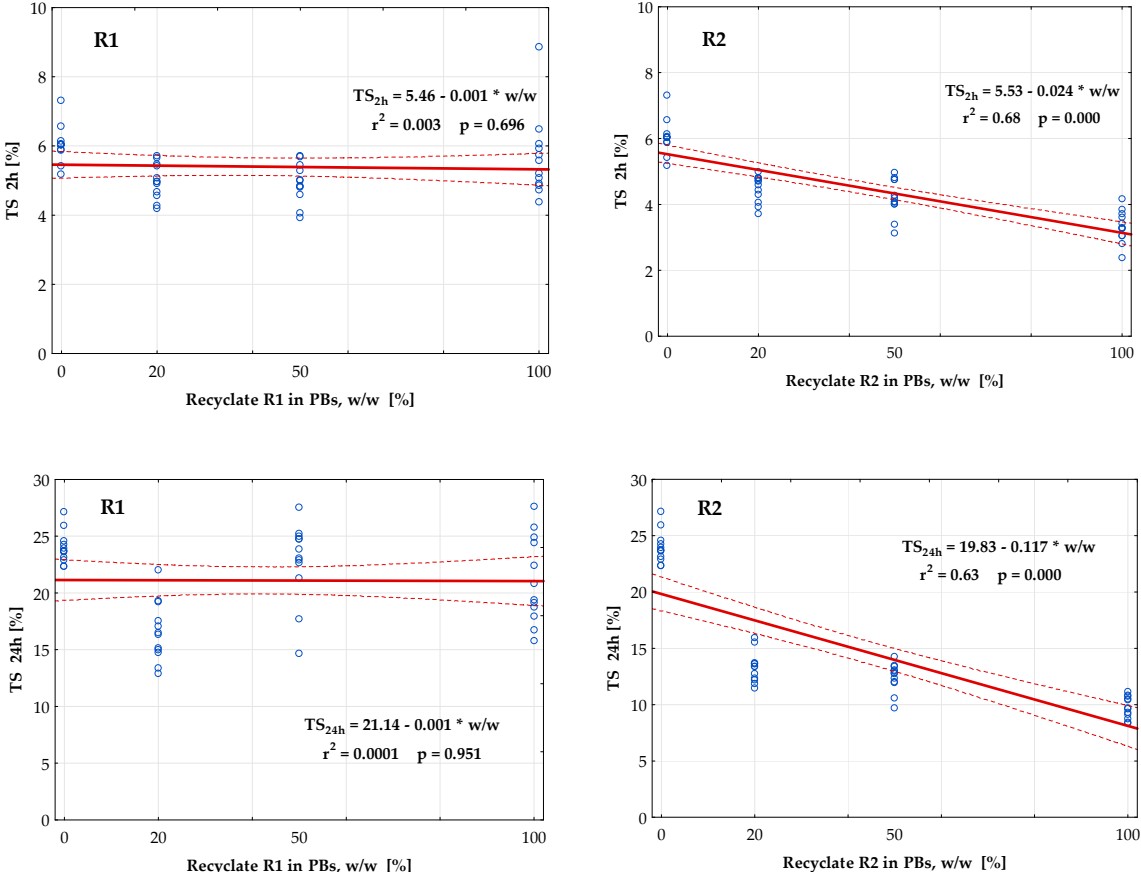

**Figure 4.** Thickness swelling (TS) after 2 and 24 h of PBs containing different types (R1 or R2) and amounts of wood recyclates.

Similar results were obtained by some other researchers, when in most cases, the moisture properties of PBs based on wood or cellulose waste improved. Azambuja et al. [37] observed the most apparent decrease in thickness swelling (TS) and water absorption (WA) in the case of PBs containing particles from recycled PBs (TS-2h, 5.77%; TS-24h, 15.16%; WA-2h, 17.32%; WA-24h, 44.41%), or particles from the mixture of wood waste "PBs, MDF boards, timber and plywood" (TS-2h, 5.46%; TS-24h, 16.12%; WA-2h, 17.40%; WA-24h, 52.94%). The control PB from pine wood particles had higher values of swelling (TS-2h, 7.49%; TS-24h, 23.16%) and also of water absorption (WA-2h, 24.46%;

WA-24h, 70.47%). However, the PBs containing the recyclate from the plywood showed worse moisture properties (TS-2h, 9.42%; TS-24h, 26.71%; WA-2h, 37.68%; WA-24h, 79.40%) than the control PB.

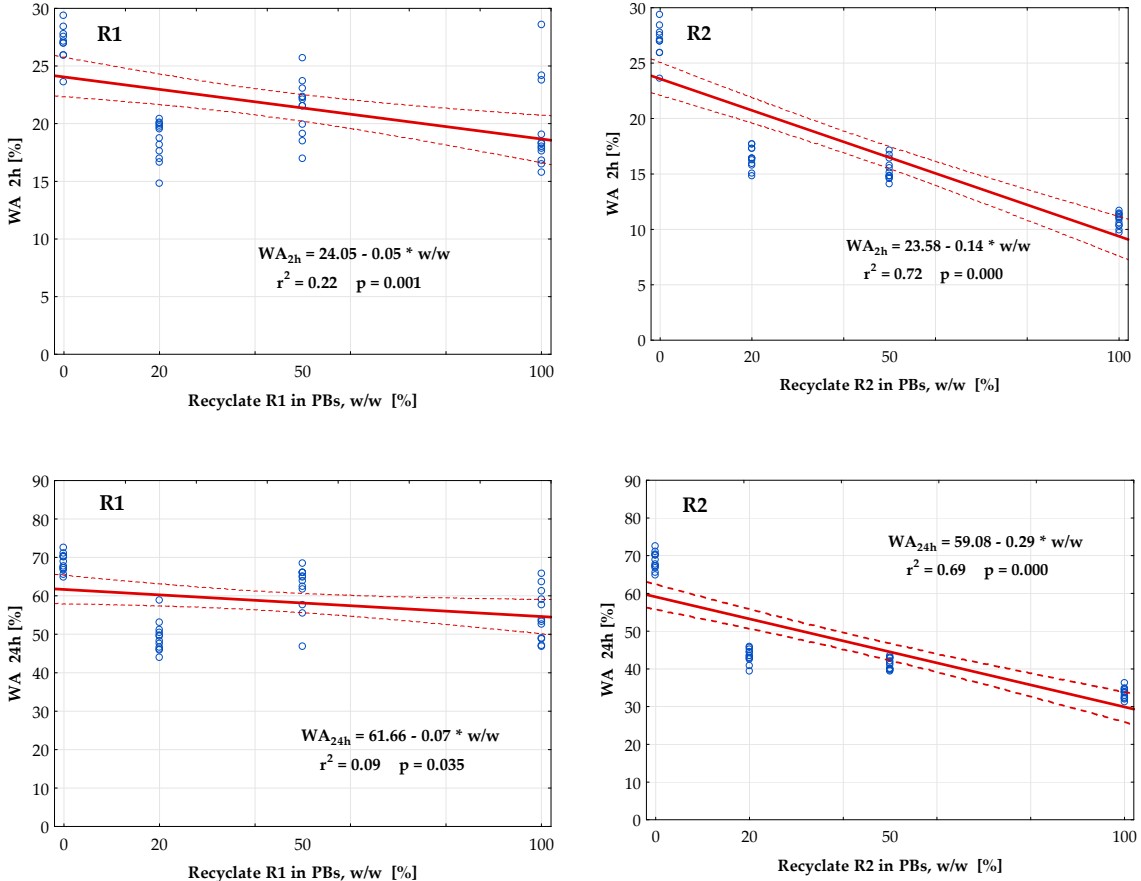

**Figure 5.** Water absorption (WA) after 2 and 24 h of PBs containing different types (R1 or R2) and amounts of wood recyclates.

Laskowska and Mamiński [36] mentioned that in the case of the recyclates from plywood bonded with UF and PF resin added to the core layer of PBs, the thickness swelling (TS-24h, 8 to 10%) in the boards manufactured from recycled plywood bonded only with UF resin was similar to the swelling in the control PB from wood particles. On the other hand, the thickness swelling of the PBs manufactured based on a waste mixture—i.e., from recycled plywood bonded with PF resin or UF resin and with the recyclate proportions of 25, 50, 75 and 100 wt.%—was 25 to 200% higher compared to the control PB bonded with UF resin. Even when the proportion of the wood particles from the recyclate of plywood bonded with PF resin was 100%, delamination of the PB was observed before testing the swelling.

Weber and Iwakiri [38] studied the moisture properties of PBs manufactured from particles of recycled MDF boards, PBs and plywood. The moisture properties of a PB with a 10 wt.% proportion of UF resin were usually better than the properties of the control PB. The best moisture properties were observed in the case of the PBs manufactured from MDF recyclate (TS-24h, 6.42%; WA-24h, 16.17%). On the contrary, the worst moisture properties were shown for PBs from recycled plywood (TS-24h, 19.80%; WA-24h, 44.33%).

The negative effect of the hydrothermal treatment (HTT) of wood particles on the moisture properties of PBs was mentioned by Lykidis and Grigoriou [17]. The effect of the primary HTT of wood particles, as well as of secondary HTTs, on the thickness swelling and water absorption of PBs was negative (TS-24h, from 37.03 to 59.11%; WA-24h, from 92.47 to 119.61%) in comparison to the control PB (TS-24h, 28.45%; WA-24h, 82.71%).

An interesting experiment was conducted by Nourbakhsh and Ashori [51]. They manufactured PBs with newspaper waste added to poplar (*Populus deltoides*) particles in amounts of 0, 25, 50 and 75 wt.%. The PBs manufactured in this way had TS-24h ranging from 12.1 to 25.9%, also with a dependence on the pressing temperature. For example, when the pressing temperature was 175 °C, the swelling thickness of the PBs with the higher proportion of newspaper was proportionally higher after 24 h: 0 wt.% → TS-24h, 12.1%; 25 wt.% → TS-24h, 14.4%; 50 wt.% → TS-24h, 17.8%; 75 wt.% → TS-24h, 21.6%.

The mechanical properties of the PBs in bending, i.e., the modulus of rupture (MOR) and the modulus of elasticity (MOE), were negatively influenced by both types of recyclates—R1 and R2 (Tables 3 and 4, Figure 6).

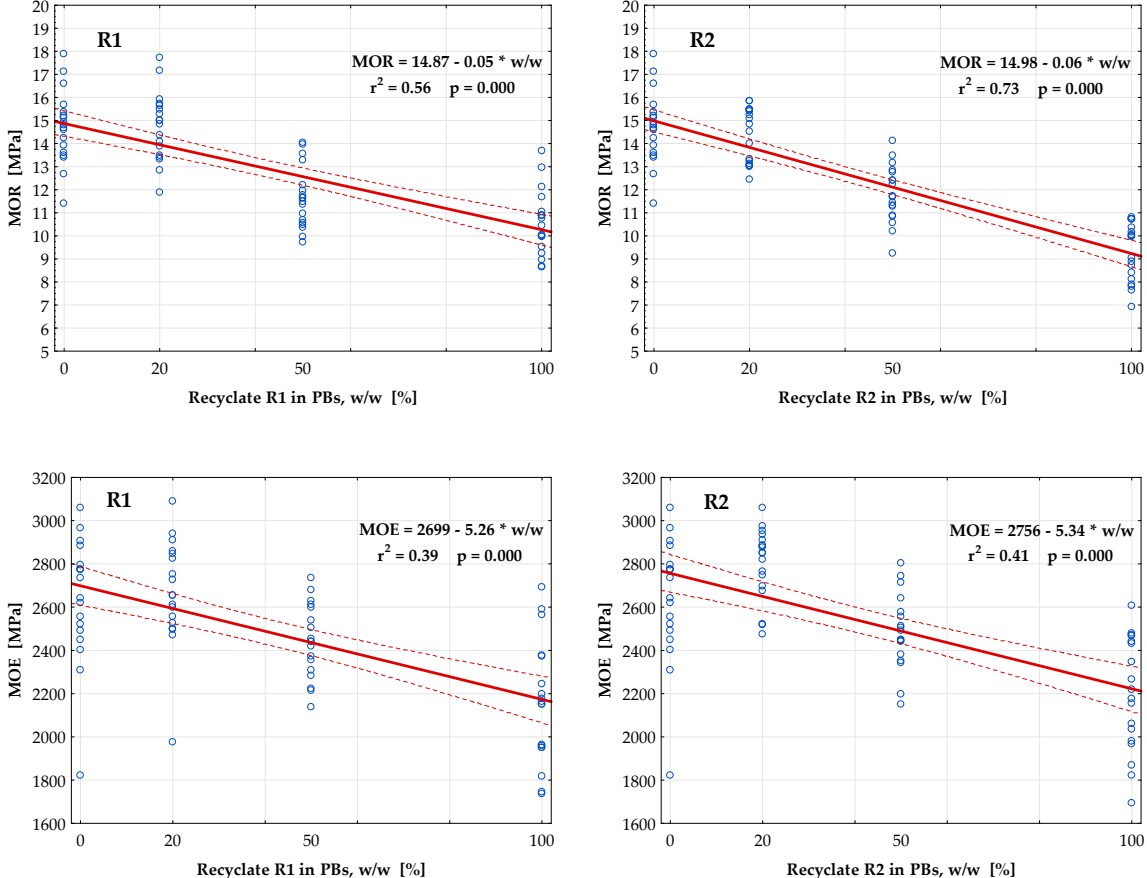

**Figure 6.** Modulus of rupture (MOR) and modulus of elasticity (MOE) of PBs containing different types (R1 or R2) and amounts of wood recyclates.

The maximal decrease in the MOR, from the 14.7 MPa of PB-C, was determined in the case of using the highest 100% amount of recyclates, i.e., by 28% to 10.6 MPa for PB-R1-100 and by 37% to 9.30 MPa for PB-R2-100. The maximal decrease in the MOE, from the 2637 MPa of PB-C, was determined similarly when the highest 100% amount of recyclates was used, i.e., by 18% to 2155 MPa for PB-R1-100 and by 17% to 2194 MPa for PB-R2-100 (Tables 3 and 4). A significantly negative effect of both wood recyclates, R1 and R2, on the bending properties of the PBs was confirmed by the coefficients of determination $r^2$ of the linear correlations "MOR or MOE = a + b × $w/w$", ranging from 0.39 for the MOE to 0.73 for the MOR (Figure 6). The mechanical properties (MOR and MOE) of the PBs based on wood recyclates fulfilled the requirements of particleboard type P2 (according to EN 312 [40]: MOR, 11 MPa; MOE, 1600 MPa) except in the MOR for the PB variants with 100% recyclates.

The internal bond (IB) of the control PB-C of 0.82 MPa decreased when there was an increase in the content of R1 or R2 recyclates in the laboratory-prepared PBs, by a maximum of about 33% to 0.55 MPa for PB-R1-100 and also for PB-R2-100 (Tables 3 and 4). A significantly negative effect of the increased amount of wood recyclates R1 and R2 on the IB of the PBs was confirmed by the coefficients of determination $r^2$ of the linear correlations "IB = a + b × *w/w*" of 0.65 and 0.70 (Figure 7). Compared to the requirements of the standard EN 312 [40], the PBs based on wood recyclates achieved, in all cases, the IB of 0.35 MPa needed for the type P2, and even 57% higher IB values for the PBs prepared from 100% recyclates.

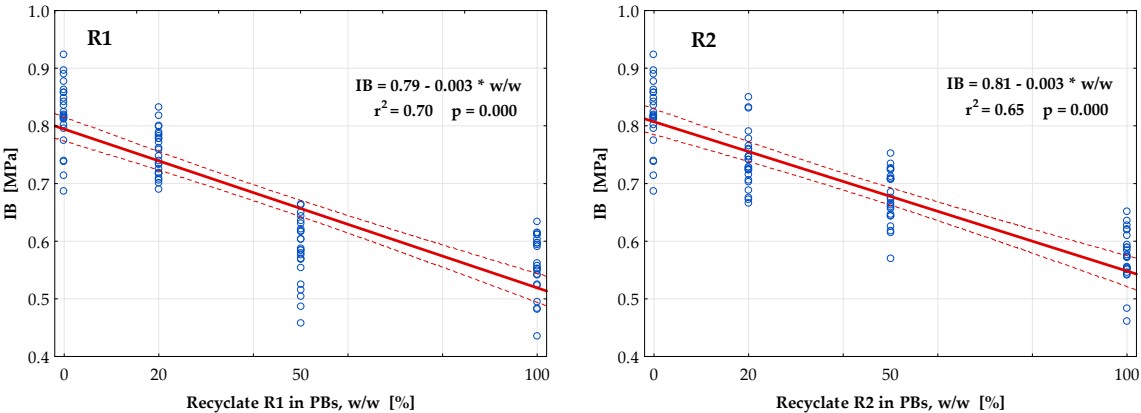

**Figure 7.** Internal bonds (IB) of PBs containing different types (R1 or R2) and amounts of wood recyclates.

In general, following several experiments—on one hand, those presented in this paper, as well as those conducted by many researchers—it can be stated that negative or, in several cases, even positive changes in the mechanical properties of PBs with wood, lingo-cellulose and cellulose recyclates depend on the type and the portion of particles in the newly manufactured composite board or on the type and portion of the adhesive used. This is confirmed by the results of the following research studies.

Azambuja et al. [37] found out that in many cases, the bending characteristics MOR and MOE of PBs with various types of wood recyclates decreased in comparison to the control PB made from pine particles (MOR, 8.00 MPa; MOE, 1378 MPa). The lowest values of the MOR and MOE, even below the requirements mentioned in the standards, were obtained in the case of PBs manufactured from waste MDF boards (MOR, 4.59 MPa; MOE, 497 MPa) and from PBs (MOR, 6.49 MPa; MOE, 1013 MPa), while low values were also observed in the case of PBs from plywood (MOR, 7.12 MPa; MOE, 1299 MPa). On the contrary, PBs manufactured from the recyclate of timber showed better bending characteristics (MOR, 9.69 MPa; MOE, 1392 MPa) compared to the control PB. This could be caused by the relatively high quality and strength of timber from coniferous trees built in trusses, ceilings and other structural parts over the decades under conditions unsuitable for decay and other deterioration processes in wood [52]. The wood recyclates also affected the IB values of the PBs in different ways—in a range from 0.18 MPa when using the waste from MDF (apparently below the standard level, 0.35 MPa) up to 0.96 MPa when using the recyclate from timber.

Laskowska and Mamiński [36] mentioned that recycled particles from plywood added to the core layer of PBs resulted in a decrease in the MOR by 13 to 91% compared to the control PB. Such a significant decrease in bending strength could be due to the less-narrow shape of wood particles from plywood and their larger area. Therefore, they would not be well bonded together with the UF resin in the newly prepared PBs.

Weber and Iwakiri [38], in the case of PBs based on various wood recyclates, determined a decrease in the MOR and MOE. For example, when using 10% UF resin, the values of the MOR ranged from 5.75 MPa for PBs based on the mixed recyclate from PBs and plywood up to 9.76 MPa for those based on the recyclate from MDF boards. The finding that in the case of the PBs from the recyclate of

MDF, the values of the MOE, 1129 MPa, were the lowest can be considered as interesting—and it was probably caused by the short, thin wood fibers from the MDF boards in prepared PBs. On the contrary, the highest values of the MOE, 1522 MPa, were observed in the case of the mixture of the waste of PBs and plywood. The decrease in IB values, in comparison to the control PB, did not appear in the case of the PBs based on the particles from MDF, PBs, plywood recyclates, and their combinations. The IB values of such prepared PBs ranged from 0.46 MPa when using the recyclate from MDF boards up to 0.56 MPa when using the recyclate from PBs and MDF boards. However, when applying a smaller 6% amount of UF resin to the PBs with wood recyclates, a decrease in all the mechanical properties was more evident. In the case of PBs based on MDF, the value of the IB was only 0.26 MPa.

The strength characteristics (MOR and IB) of the PBs were negatively affected by adding the hydrothermally modified (HTT) wood particles [17]. The value of the MOR in the PBs after the first HTT of the wood particles ranged from 12.11 to 14.24 MPa, and the value of IB, from 0.379 to 0.712 MPa. In the case of the PBs manufactured from the wood particles modified with two subsequent HTTs, the values of the MOR, 9.52 to 13.55 MPa, and of the IB, 0.177 to 0.504 MPa, were significantly lower compared to the values of the control PB (MOR, 17.15 MPa; IB, 0.938 MPa). On the other hand, the MOE values of the PBs containing HTT wood particles even partly increased—for one HTT, from 2198 to 2402 MPa, or for two HTTs, from 2379 to 2581 MPa—compared to the control PB with a MOE of 2137 MPa. This could be due to the higher toughness of wood particles after changes in their molecular structure and, again, obtaining cooled lignin macromolecules in a glassy state in the wood particles.

Nourbakhsh and Ashori [51] found out that when the newspaper/poplar wood waste ratio in PBs was increased, the mechanical properties (MOR, MOE and IB) of the boards were worse. At a pressure temperature of 175 °C, in the case of the PBs with 50% newspaper waste, the MOR was 24.7 MPa, the MOE was 2357 MPa and the IB was 0.7 MPa, or in the case of PBs with 75% newspaper waste, the MOR was 17.8 MPa, the MOE was 1858 MPa and the value of the IB was 0.57 MPa. The reference PB, prepared only from poplar particles, showed better mechanical properties (MOR, 28 MPa; MOE, 2748 MPa; IB, 1.16 MPa).

### 3.2. Biological Resistance of PBs

The biodegradation of PBs by the brown-rot fungus *Coniophora puteana* was more pronounced if the boards contained the R1 recyclate, and on the contrary, it was suppressed by the addition of the R2 recyclate (Table 5, Figures 8–10). *C. puteana* caused a 11.26% mass loss of the reference PB-C. The PB-R1-100 was characterized by the greatest mass loss, 14.86% (Δm increased by 32%), and the PB-R2-100 had the slightest mass loss, 6.26% (Δm decreased by 44%). This phenomenon is similar to the one mentioned before when evaluating a positive effect of R2 recyclate from faulty PBs on the evidently improved moisture properties of newly prepared PBs. The coefficients of determination $r^2$ of the linear correlations "Mass loss = a + b × $w/w$" were not the highest, at 0.24 or 0.33; however, on the basis of them, the above-mentioned tendencies were confirmed with sufficient significance—a more intensive decay of PBs containing the old wood "recyclate R1" and a less intensive decay of PBs containing particles covered with UF resin obtained from the failed PBs, "recyclate R2" (Figure 10).

The increased fungal attack of PBs containing Rl recyclate can be explained by their composition, e.g., (a) the presence of old furniture produced from beech particles over 50–60 years of the 20th century in the "Bukas" PBs from the company Bučina Zvolen, Czechoslovakia; according to EN-350 [53], the decay resistance of beech wood is 5, "non-durable", while that of spruce wood is better, 4, "less durable", (b) the presence of HB and MDF boards containing defibrated fine wood fibers, which are easily accessible to hydrolase and other fungal enzymes. The decreased decay of PBs containing R2 recyclate can be attributed to the higher amount of cured UF resins on the surfaces of wood particles.

**Table 5.** Biological resistance of PBs containing recyclates R1 or R2—valued on the basis of mass losses (Δm) caused by the brown-rot fungus *Coniophora puteana.*

| Recyclate Type in PB | Mass Loss of PB Caused by *C. puteana* (%) Recyclates R1 or R2 in PB *w/w* (%) | | | |
|---|---|---|---|---|
| | **0** | **20** | **50** | **100** |
| R1 | 11.26 (2.57) | 12.43 (2.14) | 12.81 (1.76) | 14.86 (3.30) |
| R2 | 11.26 (2.57) | 7.52 (1.47) | 6.69 (2.40) | 6.26 (1.61) |

Notes: Mean values: mass loss due to *C. puteana* from 6 samples. Standard deviations are in the parentheses. R1 = particles from mixture of several recycled wood products; R2 = particles from recycled faulty PBs.

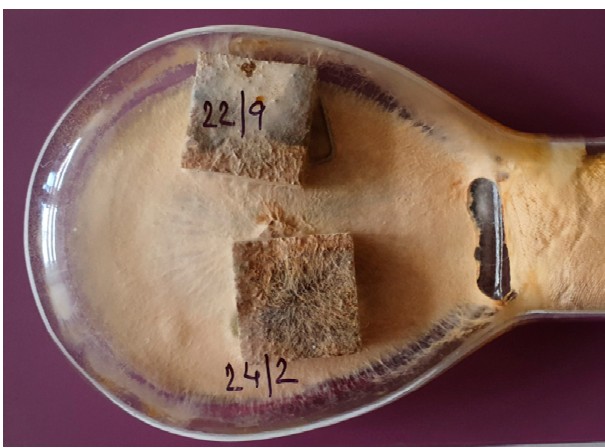

**Figure 8.** Growth of the surface mycelia of the brown-rot fungus *Coniophora puteana* on the top surfaces of PB-20-R1 (it contains 20% particles from a mixture of recycled wood products R1 and 80% particles from fresh spruce wood) after 16 weeks of the decay test in Kolle flasks.

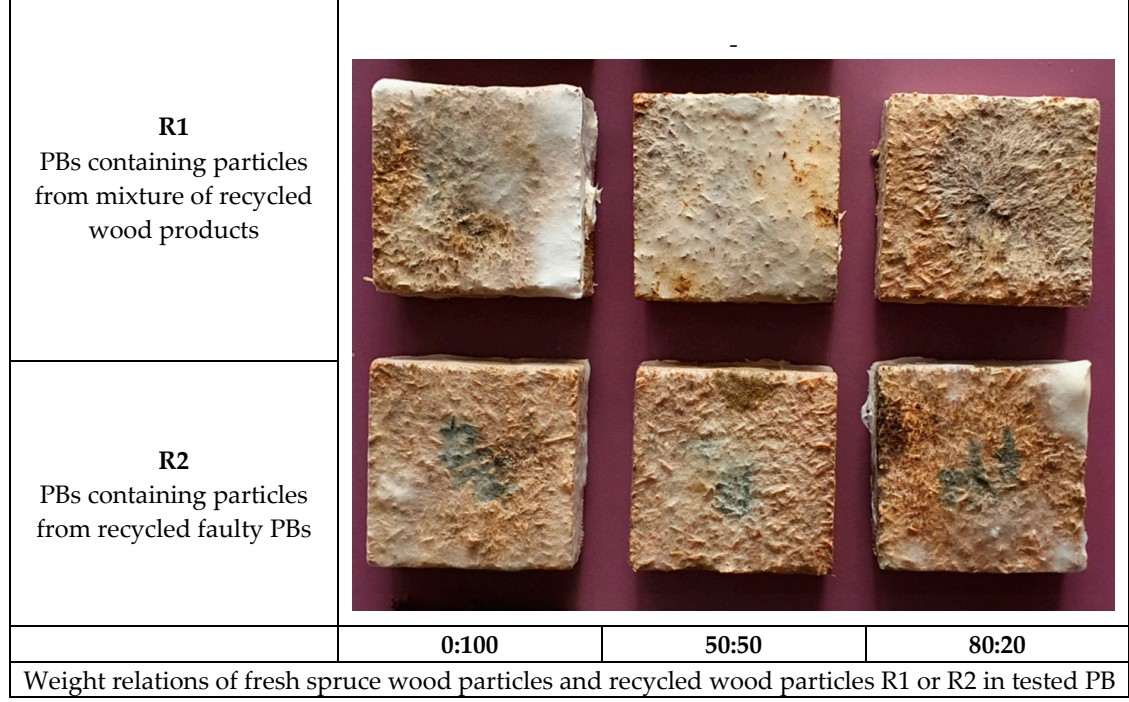

| R1 PBs containing particles from mixture of recycled wood products | |
|---|---|
| R2 PBs containing particles from recycled faulty PBs | |
| | **0:100** **50:50** **80:20** |
| Weight relations of fresh spruce wood particles and recycled wood particles R1 or R2 in tested PB | |

**Figure 9.** Growth of the surface mycelia of the brown-rot fungus *Coniophora puteana* on the top surfaces of PBs containing particles from recycled wood after 16 weeks of the decay test.

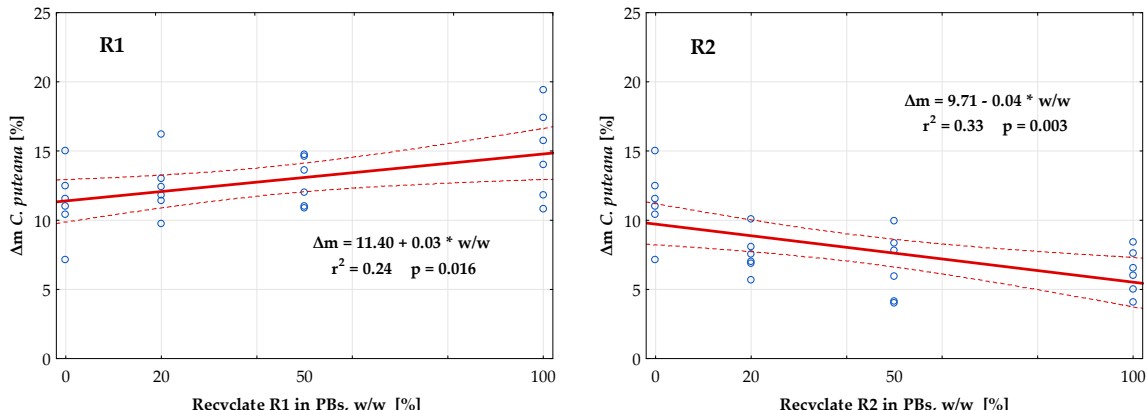

**Figure 10.** Biological resistance of PBs containing different types (R1 or R2) and amounts of wood recyclates to the brown-rot fungus *Coniophora puteana*, determined by mass losses "Δm" (mycological test according to ENV 12038 [50]).

The interactive effect of the moisture properties and decay resistance on panel composites is also reported by other studies [54–56], in which a higher water absorption by wood particles resulted in a more intensive decay in such composites. The biological resistance of panel products made from waste biomaterials to wood-decaying fungi is also influenced by the type of resin used [55,57] and biocides [58,59] and other additives [60] present in various wood composite types.

## 4. Conclusions

- The thickness swelling (TS) and the water absorption (WA) of PBs prepared from wood recyclates decreased more apparently when using R2 particles prepared from faulty UF-bonded PBs—with a decrease in TS after 24 h by 59% and in WA after 24 h by 51%. This was probably due to the presence of cured UF resin on the surfaces of such recycled wood particles. Adding the R1 particles from the old wood products did not apparently affect the moisture properties of the PBs.
- The modulus of rupture (MOR) in bending, the modulus of elasticity (MOE) in bending, and the internal bond (IB) of the PBs were negatively affected by a larger amount of R1 and R2 recyclates in them. The most apparent decrease in all the tested mechanical properties was shown for the PBs prepared only from recyclates—containing 100% R1 or R2—i.e., there was a decrease in the MOR from 14.7 to 10.6 or 9.30 MPa, a decrease in the MOE from 2637 to 2155 or 2194 MPa, and a decrease in the IB from 0.82 to 0.55 MPa.
- The biological resistance of PBs to the brown-rot fungus *Coniophora puteana* decreased in the presence of the R1 recyclate, from the old wood products. On the contrary, the R2 recyclate, from the faulty PBs, evidently suppressed the decay activity of *C. puteana* in newly prepared PBs. The R2 recyclate had, in this situation, a similar effect as was determined in the case of the moisture properties of the PBs—probably due to the presence of some portion of cured UF resin on its surfaces.
- Adding the suitable types of wood recyclates to PBs is important in terms of economy and the environment as well. Generally, PBs containing wood recyclates could be used for less or more stressed exposures, considering the used type and amount of recyclate content.
- The three-layered UF resin-bonded PBs based on wood recyclates showed sufficient mechanical properties, complying with EN 312 [40], and fulfilled the requirements for particleboard type P2, except in the MOR for PB variants with 100% recyclates.

**Author Contributions:** Conceptualization, J.I., Z.V. and L.R.; methodology, J.I., Z.V. and L.R.; software, J.I. and Z.V.; validation, J.I., Z.V. and L.R.; formal analysis, J.I. and L.R.; investigation, J.I., Z.V. and L.R.; resources, J.I., Z.V. and L.R.; data curation, J.I., Z.V. and L.R.; writing—original draft preparation, J.I., Z.V. and L.R.; writing—review and editing, J.I. and L.R.; visualization, J.I. and Z.V.; supervision, L.R.; project administration, J.I. and L.R.; funding acquisition, L.R. All authors have read and agreed to the published version of the manuscript.

**Funding:** This work was supported by the Slovak Research and Development Agency under the contract no. APVV-17-0583.

**Acknowledgments:** We would like to thank, for providing some of the materials for this research, the companies Kronospan, s.r.o., Zvolen, Slovakia, and Kronospan CR, spol. s r.o., Jihlava, Czech Republic.

**Conflicts of Interest:** The authors declare no conflict of interest.

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
