# Peer review of "Particleboards from Recycled Wood"

_forests, doi:10.3390/f11111166_

Round 1
Reviewer 1 Report
The manuscript concerns the use of recycled wood in the manufacture of particleboard.
The manuscript is interesting, correctly written, references well selected.
The references/comparisons with previous works in the Results and Discussion section are especially valuable.
Below is a list of my comments:
-, line 11 and later, Abbreviations should be defined in parentheses the first time they appear in the abstract, main text, and in figure or table captions and used consistently thereafter.
- line 37, “43.9 mil. m3, abbreviation “mil.” is not clear-cut, I suggest using the full form or notation “106”.
- lines 121 and 200, please correct the unit.
- lines 179 and 185, it should be “m0” instead of “mo” (as in formula), please correct.
- line 310, it should be “MOR and MOE”, please check.
- References, Please check and unify the references, e.g. [14] - the patent number is incorrect, [18] - the DOI number is incorrect, [21] - A1 means that it is a patent application.
Author Response
Point 1 - line 11 and later, Abbreviations should be defined in parentheses the first time they appear in the abstract, main text, and in figure or table captions and used consistently thereafter.
Response 1: - yes we agree, we have incorporated this comment in the individual parts of the article
Point 2: - line 37, “43.9 mil. m3, abbreviation “mil.” is not clear-cut, I suggest using the full form or notation “106”.
Response 2: - yes it is modified in the article, now is used the form “43.9x106 m3”
Point 3: - lines 121 and 200, please correct the unit.
Response 3: In line 121 is 650 ± 10 kg.m-3, it shows the density relating to the range of densities of the particleboards produced; and in line 200 – it shows the range of average values of densities for the individual variants of the particleboard produced “649 and 658 kg.m-3”
- the unit is modified, kg · m-3
Point 4: - lines 179 and 185, it should be “m0” instead of “mo” (as in formula), please correct.
Response 4: - yes, it is modified
Point 5: - line 310, it should be “MOR and MOE”, please check.
Response 5: yes, it is modified
Point 6: - References, Please check and unify the references, e.g. [14] - the patent number is incorrect, [18] - the DOI number is incorrect, [21] - A1 means that it is a patent application.
Response 6: yes, it is modified

Reviewer 2 Report
It is a very interesting article that combines own research with extensive literature study in the field of the possible use of wooden waste in the production of particle boards.
Research into the use of wood waste is enormously important for extending the use cascade. At the same time, it is difficult to produce a standard-compliant product from various wood wastes. This article shows first laboratory scale tests, which show a very detailed approach regarding the mixing ratios (natural product : wood waste) and are evaluated in a practice-oriented way, because they are tested and evaluated according to the material standard. The uniform approach with regard to temperature, pressing time and pressure is useful in the context of such research in order to ensure comparability of the results and not to get lost in too many variables.
The results are clear, comprehensible and stated in statistical relation. The research of comparative studies has been extensive, possibly reproduced in somewhat too much detail, which somewhat disturbs the simple flow of reading by too many comparative figures. However, the focus of scientific texts should be on the information content rather than the simple reading flow.
Author Response
Point 1
It is a very interesting article that combines own research with extensive literature study in the field of the possible use of wooden waste in the production of particle boards.
Research into the use of wood waste is enormously important for extending the use cascade. At the same time, it is difficult to produce a standard-compliant product from various wood wastes. This article shows first laboratory scale tests, which show a very detailed approach regarding the mixing ratios (natural product: wood waste) and are evaluated in a practice-oriented way, because they are tested and evaluated according to the material standard. The uniform approach with regard to temperature, pressing time and pressure is useful in the context of such research in order to ensure comparability of the results and not to get lost in too many variables.
The results are clear, comprehensible and stated in statistical relation. The research of comparative studies has been extensive, possibly reproduced in somewhat too much detail, which somewhat disturbs the simple flow of reading by too many comparative figures. However, the focus of scientific texts should be on the information content rather than the simple reading flow.
Response 1 – yes, we agree with your view on the form of presentation of several results in the article, so that some detailed results can disturb the simply flow of reading, however, we wanted to present the results in as condensed and concise a form as possible.

Reviewer 3 Report
Review comments The topic of the manuscript is of great interest and quite novel, dealing with the recycling of wood in particleboards. Although, the manuscript is well prepared, I would like to make the following recommendations for its improvement. In line 115- Concerning the Paraffin emulsion, the authors do not refer in what percentage was added. in line 129- you should refer the exact duration of pressure application. In line 129- I believe that you should change it and correct it as 8 s/mm in the text. In line167- as it is obvious in figure 2, the perimetric frame of the PBs has not been removed, which usually does not have the desired structure of the rest of the board. In line 171- you do not refer why did you use the specific fungus C.puteana. Did you use correction flasks without mycelia to check the interaction of medium with the samples? In line 409- please, provide a justification on that.Author Response
Point 1 - In line 115- Concerning the Paraffin emulsion, the authors do not refer in what percentage was added.
Response 1: “Paraffin emulsion, with 35% of dry mass, was applied on the surface and core particles in the amount of 0.6% and 0.7%, respectively.”
Point 2: - in line 129- you should refer the exact duration of pressure application
Response 2: The exact pressing time is shown in the Figure 1 in the pressing diagram, the pressing time in the text is given as a parameter pressing factor 8 s/mm.
Point 3: - In line 129- I believe that you should change it and correct it as 8 s/mm in the text.
Response 3: - yes, it is modified
Point 4: - In line167- as it is obvious in figure 2, the perimetric frame of the PBs has not been removed, which usually does not have the desired structure of the rest of the board.
Response 4: In Figure 2. - the bold line represents only a symbolic cutting - the board was of course cut around the perimeter.
Point 5: - In line 171- you do not refer why did you use the specific fungus C.puteana.
Response 5: The brown-rot fungus Coniophora puteana is a typical domestic fungus which often occurs in interior of buildings. Since we produced particleboard type P2 for the indoor environment that it is why we used this fungus (this fungus is recommended for mycological laboratory tests by European standards, as well).
Point 6: - Did you use correction flasks without mycelia to check the interaction of medium with the samples? In line 409- please, provide a justification on that.
Response 6: No, we did not use. So we know that the achieved results can be influenced by some error resulting from a different release of VOCs from the different types of PBs. But we think, that the tendencies related to a higher mass loss for PBs with R1 recyclate, and to a lower mass loss for PBs with R2 recyclate could not be more importantly influenced by this.

Reviewer 4 Report
The manuscript describes the preparation of particleboards from recovered wood. The research presented is rather applied but relevant from an industrial perspective. There are several articles about particleboards and recovered wood in the literature; this manuscript adds some scientific news.
My main concern is that it is difficult to reproduce the study for another research group.
"mixture of several recycled wood products made from HDF boards, MDF 99 boards, PBs, pallets, old furniture, etc. (R1), " - how can someone reproduce these experiments from this description?
"recycled faulty PBs bonded with UF resin (R2)." - Is it possible to describe this better? Faulty in what way?
Some other comments:
Row 126: Particle mat was pre-pressed at a laboratory temperature - This is unclear
3.2 Biological degradation
This paragraph could be improved with a better and more deep discusion. Why is there a difference between R1 and R2? A better and more clear discusion is needed.
"On the contrary, their attack by C. puteana was 363 suppressed by addition of the R2 recyclate (Table 5, Figure 10)." - what does their refer to in this sentense?
Author Response
Point 1 - "mixture of several recycled wood products made from HDF boards, MDF 99 boards, PBs, pallets, old furniture, etc. (R1), " - how can someone reproduce these experiments from this description?
Response 1: PBs-R1 – i.e., the R1 recyclate was prepared from mixture of several recycled wood products, approximately in the following percentage content: 35% MDF and HB boards, 30% PBs, 20% pallets from spruce wood, 15% old furniture - including old types of PBs prepared from beech particles and treated with beech or other hardwood veneers.
Point 2: "recycled faulty PBs bonded with UF resin (R2)." - Is it possible to describe this better? Faulty in what way?
Response 2: The faulty PBs were particleboards prepared in the company Kronospan Zvolen, which immediately after production did not meet the technical requirements for the PB type P2 and therefore were discarded and subsequently used for recycling.
Point 3: Row 126: Particle mat was pre-pressed at a laboratory temperature - This is unclear
Response 3: - yes, it is modified – Particle mat was cold pre-pressed at a temperature environment…
Point 4: 3.2 Biological degradation This paragraph could be improved with a better and more deep discusion. Why is there a difference between R1 and R2? A better and more clear discusion is needed.
Response 4: - yes, it is modified.
An increased fungal attack of PBs with content of Rl recyclate can be explained by their composition, e.g., a) presence of old furniture from “Bukas” PBs produced from beech particles in 50-60-years of the 20-century in the company Bučina Zvolen, Czechoslovakia, as by EN-350 [53], a decay resistance of beech wood is 5 “non-durable” while of spruce wood is better 4 “less durable”; b) presence of HB and MDF boards containing defibrated fine wood fibers which are easily accessible to hydrolase and other fungal enzymes, and on the other hand, comparing to PBs with a lower portion of synthetic resins which are barrier against transport of enzymes. A decreased decay of PBs containing R2 recyclate can be attributed to a higher amount of cured UF resins on surfaces of wood particles.
Point 5: "On the contrary, their attack by C. puteana was 363 suppressed by addition of the R2 recyclate (Table 5, Figure 10)." - what does their refer to in this sentense?
Response 5: - Yes, the form of sentence was corrected.

Round 2
Reviewer 4 Report
The manuscript can now be published.